# Diameter-dependent phase selectivity in 1D-confined tungsten phosphides

Gangtae Jin [1,7], Christian D. Multunas[2,7], James L. Hart [3,7], Mehrdad T. Kiani [3], Nghiep Khoan Duong [4], Quynh P. Sam[3], Han Wang[3], Yeryun Cheon [4], David J. Hynek[5], Hyeuk Jin Han [6] ✉, Ravishankar Sundararaman [2] ✉ & Judy J. Cha [3] ✉

Topological materials confined in 1D can transform computing technologies, such as 1D topological semimetals for nanoscale interconnects and 1D topological superconductors for fault-tolerant quantum computing. As such, understanding crystallization of 1D-confined topological materials is critical. Here, we demonstrate 1D template-assisted nanowire synthesis where we observe diameter-dependent phase selectivity for tungsten phosphides. A phase bifurcation occurs to produce tungsten monophosphide and tungsten diphosphide at the cross-over nanowire diameter regime of 35–70 nm. Four-dimensional scanning transmission electron microscopy is used to identify the two phases and to map crystallographic orientations of grains at a few nm resolution. The 1D-confined phase selectivity is attributed to the minimization of the total surface energy, which depends on the nanowire diameter and chemical potentials of precursors. Theoretical calculations are carried out to construct the diameter-dependent phase diagram, which agrees with experimental observations. Our findings suggest a crystallization route to stabilize topological materials confined in 1D.

Nanostructured transition-metal phosphides (TMPs) are a promising material platform for energy storage, catalysis, and photonics[1–6]. A subset of TMPs possess topologically non-trivial electronic band structures: group V phosphides such as niobium phosphide and tantalum phosphide are Weyl semimetals[7–9], and group VI phosphides such as molybdenum monophosphide (MoP) and tungsten monophosphide (WP) are topological metals that exhibit high conductivity and high carrier density, along with the topologically protected fermions[10–13]. Additional interesting properties for WP include superconductivity with a small electron-phonon coupling strength[14–17] and multiple semi-Dirac-like points near the Fermi level[17]. Tungsten diphosphide ($WP_2$) is another topological TMP that exhibits high magnetoresistance ($3 \times 10^5$%) due to compensated

semimetal characteristics and large suppression of backscattering which can be attributed to the robust topological phase with two neighboring Weyl points[18–20]. These TMP topological semimetals[8–20] are an emerging class of promising nanoscale interconnect materials, especially when confined in 1D, which can potentially deliver the desired dimensional scaling of decreasing resistivity with decreasing dimensions, arising from the topological surface states and suppressed electron backscattering[21–24]. An anisotropic conductor within a 1D framework is also a promising interconnect metal if the high fermi velocity direction is oriented along the length of the 1D wire such that surface scattering is greatly suppressed[25]. In the context of classical expression, the resistivity of metal wires increases as their width decreases below the bulk electron mean free path ($\lambda$). This

[1]Department of Electronic Engineering, Gachon University, Seongnam 13120, South Korea. [2]Department of Materials Science and Engineering, Rensselaer Polytechnic Institute, Troy, NY 12180, USA. [3]Department of Materials Science and Engineering, Cornell University, Ithaca, NY 14850, USA. [4]Department of Physics, Cornell University, Ithaca, NY 14850, USA. [5]Department of Mechanical Engineering and Materials Science, Yale University, New Haven, CT 06511, USA. [6]Department of Environment and Energy Engineering, Sungshin Women's University, Seoul 01133, South Korea. [7]These authors contributed equally: Gangtae Jin, Christian D. Multunas, James L. Hart. ✉e-mail: hyeukjin.han@sungshin.ac.kr; sundar@rpi.edu; judy.cha@cornell.edu

increase in resistivity is due to contributions from surface and grain-boundary scattering[25–28].

Despite the promises for low-resistance nanoscaled interconnects, controlled synthesis of nanostructured topological semimetals has been under-investigated. Nevertheless, precision synthesis of 1D-confined TMPs is essential for the realization of energy-efficient computing technologies based on their emergent transport phenomena.

For crystallization in 1D, the free-energy landscape, which underlies the kinetics and thermodynamics of the crystallization process, is significantly affected by the nanoscale confinement[29–31]. At large surface-to-volume ratios, surface energy of the growth products can dominate the crystallization pathway: a metastable phase might be preferred over a stable phase if the surface energy of the metastable phase is lower. Thus, nanoscale confinement can be exploited to control phase stability and synthesis pathways[32,33]. A 2D-confined template is commonly used to achieve transition-metal nitrides and transition-metal phosphides in recent reports. This method ensures a homogeneous phase across the unconventional 2D structures, regardless of the template's thickness, e.g., $MoS_2$, $TiS_2$, or $WS_2$[13,34]. For these crystallization in confined dimensions, atomistic understanding can be further developed to achieve phase selectivity via geometric confinement.

Here, we developed a 1D-confined synthesis of topological metal WP and $WP_2$ via 1D template-assisted transformations. With decreasing diameter, we observe crystallization pathways, resulting in distinct phases. The size effects on phase stability of tungsten phosphides were studied using four-dimensional scanning transmission electron microscopy (4D-STEM) and automated crystal orientation mapping

(ACOM) on 4D STEM datasets[35,36]. Diameter-dependent phase selectivity of WP or $WP_2$ is attributed to the surface-energy differences of the synthesized nanowires. A diameter-dependent phase diagram was constructed by density-functional-theory (DFT) calculations, which support the observed WP and α-$WP_2$. For the interconnect applications, we measured resistivity of the synthesized nanowires and show that polycrystalline 1D-WP should have minimized surface electron scattering due to its small mean free path of 3.15 nm. Our findings demonstrate that diameter-dependent crystallization is a viable synthesis route for 1D topological semimetals.

## Results

### Band structure and synthesis of 1D tungsten phosphide

DFT calculations were performed to obtain electronic band structures of WP and $WP_2$. WP has an orthorombic crystal structure (Fig. 1a) with lattice parameters $a = 0.327$ nm, $b = 0.576$ nm, $c = 0.627$ nm[17]. To represent nanoscale WP with large surface-to-volume ratios, we calculated a surface-weighted band structure of an 8-unit cell thick WP slab that was terminated with the lowest surface energy crystal planes of (011) (Fig. 1b). Monoclinic α-$WP_2$ with lattice parameters $a = 0.848$ nm, $b = 0.319$ nm, $c = 0.748$ nm is one of the stable W-P compounds at room temperature (Fig. 1c). α-$WP_2$ is topologically trivial unlike β-$WP_2$ that has Weyl nodes[37]. The surface-weighted band structure of a monoclinic 11-unit cell thick α-$WP_2$ slab terminated with (110) crystal planes is shown in Fig. 1d. Band structure calculations show both are metals.

Highly anisotropic WP nanostructures were synthesized via 1D-confined transformation by converting $WO_2$ nanowires to WP nanowires, as illustrated in Fig. 1e. First, we grew 1D-$WO_2$ on the c-sapphire

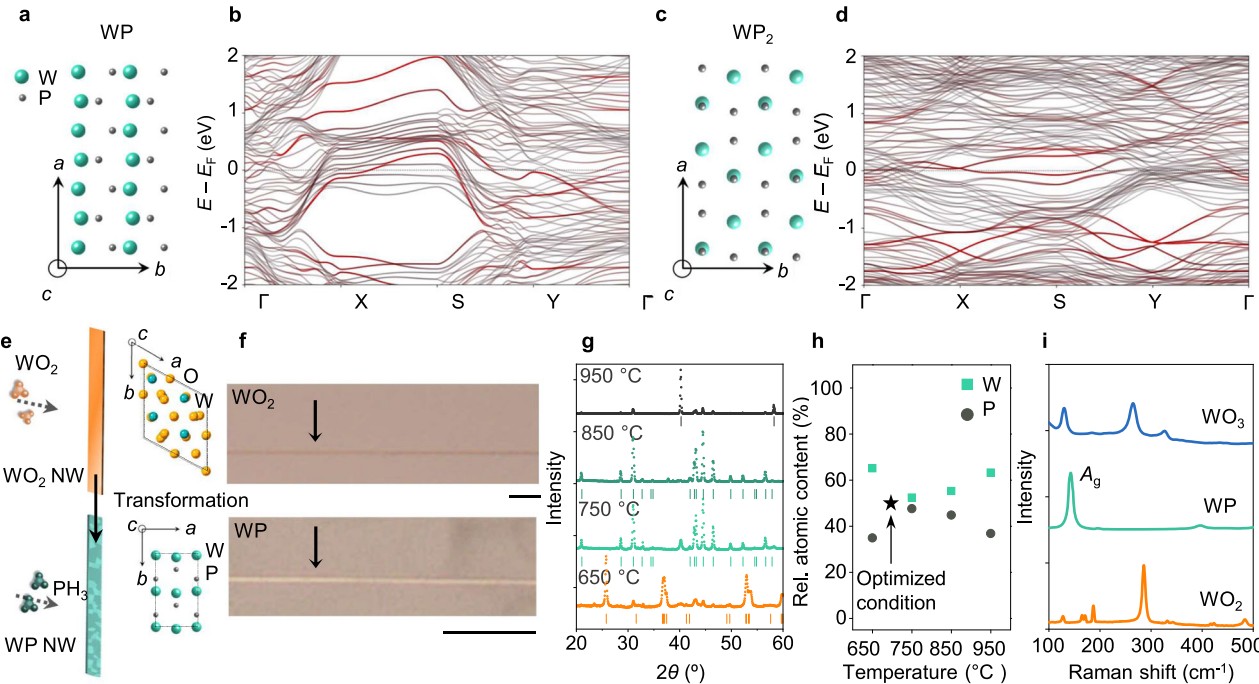

**Fig. 1 | Electron band structures and template-assisted conversion of WP and $WP_2$.** Crystal structure (**a**) and calculated surface-weighted electronic band structure (**b**) of a WP slab, terminated with the lowest surface energy plane (011). Surface states from the terminated crystal planes are colored in red. Note that these surface states do not represent topologically protected surface states. Crystal structure (**c**) and calculated surface-weighted electronic band structure (**d**) of a α-$WP_2$ slab, terminated with the lowest surface energy plane (110). Surface states from the terminated crystal planes are colored in red. We have chosen the same high-symmetry path for orthorhombic WP and monoclinic $WP_2$ since these band structures represent slabs rather than bulk structures, and the slabs have identical

surface unit cells. **e** Schematics of geometrically confined transformation from $WO_2$ to WP with corresponding crystal structures of $WO_2$ and WP. **f** Optical microscope images of $WO_2$ template grown on c-cut sapphire and transformed 1D-WP. scale bars: 10 μm. **g** Powder X-ray diffraction spectra of growth products as a function of the conversion temperature (orange, 650 °C; light green, 750 °C; dark green, 850 °C; black, 950 °C). **h** Relative atomic content of $W_xP_{1-x}$ (0.5 < x < 0.7) growth products, collected from SEM-EDS signals at 4 different conversion temperatures. **i** Raman spectra of as grown 1D-$WO_2$ (orange), transformed 1D-WP (light green), and $WO_3$ (blue), respectively.

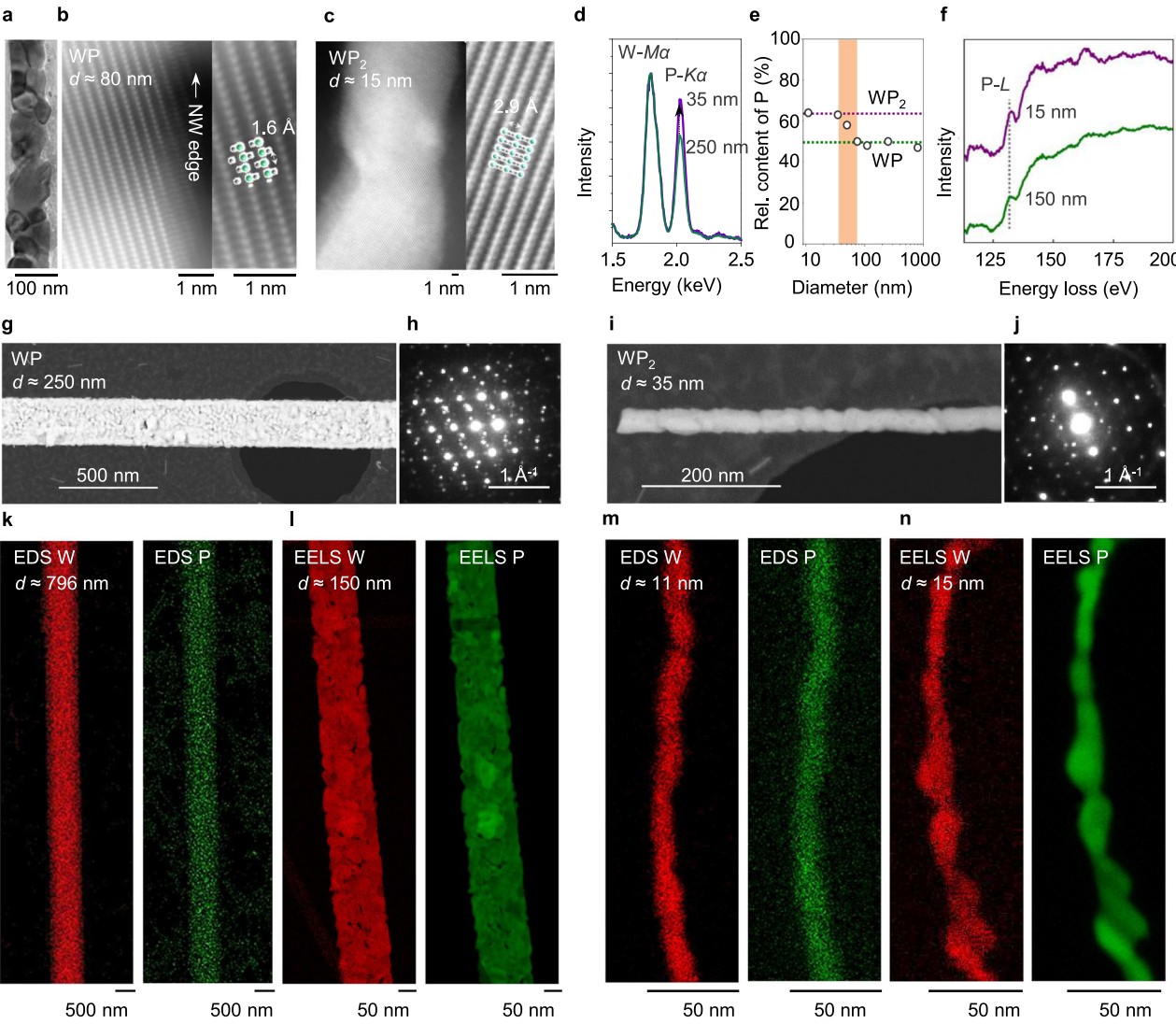

**Fig. 2 | Structural characterization of 1D-confined WP$_{x\,(x = 1\ or\ 2)}$ with varying diameter. a** Low-magnification TEM image of 1D-confined WP. Atomic-resolution HAADF-STEM and magnified images of WP (**b**) and WP$_2$ (**c**) obtained from a single grain of the polycrystalline WP and WP$_2$ nanowires. **d** STEM-EDS spectra acquired from representative 1D-confined WP$_{x\,(x = 1\ or\ 2)}$ (green, 250 nm in diameter; purple, 35 nm in diameter). **e** Relative atomic content of P with varying diameter, collected from TEM-EDS signals of 7 different samples. Cross-over regime is 35–70 nm. **f** EELS P L-edge of 1D-confined WP$_{x\,(x = 1\ or\ 2)}$ (green, 150 nm in diameter; purple, 15 nm in diameter). HAADF-STEM images of 1D-confined WP (**g**) with the diffraction pattern of dominant zone axis of [131] (**h**) and WP$_2$ (**i**) with the corresponding diffraction pattern of zone axis of [233] (**j**). False-color STEM-EDS maps (W-L in red, P-K in green) (**k**) and EELS maps (W-O in red, P-L in green) (**l**) obtained from two WP nanowires. False-color STEM-EDS maps (W-L in red, P-K in green) (**m**) and EELS maps (W-O in red, P-L in green) (**n**) collected from two WP$_2$ nanowires. The W:P ratio of the WP nanowire (diameter ≈ 796 nm) shown in (**k**) is 0.53:0.47 and the ratio of the WP$_2$ nanowire (diameter ≈ 11 nm) shown in (**m**) is 0.36:0.64.

substrates with a miscut angle of 1° along the a-axis of sapphire (⟨11–20⟩)[38,39] by vapor transport synthesis from WO$_3$ powder precursors. As-grown 1D-WO$_2$ templates (monoclinic, P2$_1$/c space group, a = 0.576 nm, b = 0.484 nm, c = 0.580 nm) were transformed to orthorhombic WP via phosphorization using PH$_3$ gas, produced from the thermal decomposition of NaH$_2$PO$_2$·H$_2$O at 700 °C (see Methods and Supplementary Fig. 1 for growth aspects). The transformed WP exhibits a highly directional crystalline form with typical widths of 100–500 nm and lengths of 10–300 μm (aspect ratio of ≈100) as shown in Fig. 1f and Supplementary Fig. 2. The conversion from WO$_2$ to WP is accompanied by the volume decrease from the unit-cell volume of 149.31 Å$^3$ for WO$_2$ to 117.90 Å$^3$ for WP due to the change in crystal symmetry and lattice parameters. The WO$_2$ to WP transformation is apparent by the color change in converted crystals, but the original morphologies of the WO$_2$ nanowires were preserved in converted WP.

The conversion temperature was optimized by analyzing X-ray diffraction (XRD) and energy-dispersive X-ray spectroscopy (EDS) data (Fig. 1g, h). At 650 °C, the powder product still contained WO$_2$ diffraction peaks and excess W with an elemental ratio of W:P = 1.86:1, which suggest incomplete conversion to WP. At higher temperatures (>750 °C), XRD patterns mainly indicate the orthorhombic WP without the formation of other W-P phases, such as α-WP$_2$, β-WP$_2$, WP$_3$, or W$_3$P. However, the stoichiometry of converted WP deviated from the expected W:P ratio of 1:1. With increasing conversion temperature, the W:P ratio was 1.1:1 at 750 °C, 1.2:1 at 850 °C, and 1.7:1 at 950 °C. From the XRD and EDS analysis, the conversion temperature was set to 700 °C to achieve the W:P ratio of 1:1 (magnified XRD spectra in Supplementary Fig. 3). SEM-EDX analysis of individual wires are shown in Supplementary Fig. 4. To verify the formation of WP and absence of any residual tungsten oxides, we also obtained the Raman spectra (excitation 532 nm) of initial WO$_2$ templates and final WP nanostructures converted at 700 °C (Fig. 1i and Raman map in Supplementary Fig. 5). The Raman spectrum of the WO$_2$ template (orange) is in agreement with previous studies of 1D WO$_2$[40]. The

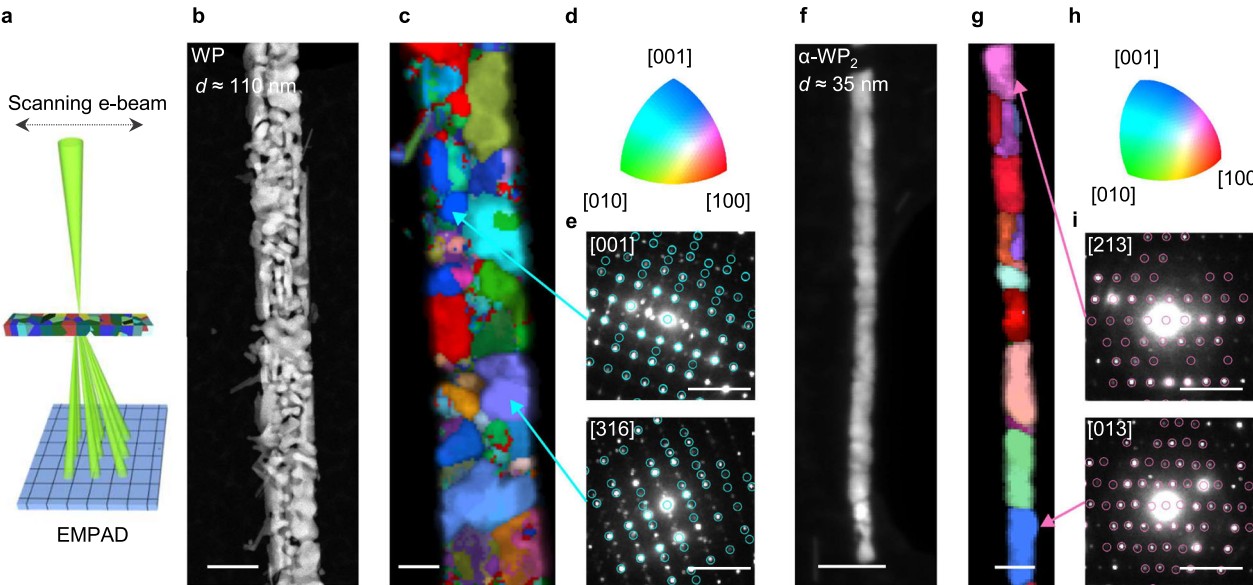

**Fig. 3 | 4D STEM and grain orientation mapping of WP and α-WP₂. a** Schematic for 4D-STEM of 1D-confined $WP_{x\ (x=1\ or\ 2)}$ on EMPAD. HAADF-STEM image (**b**) and grain orientation map (**c**) of 1D-confined orthorhombic WP. The orientation map only shows a sub-section of the wire. Each color represents a crystal plane corresponding to the color-coded inverse pole figure shown in (**d**). Inverse pole figure for WP (**d**) and electron diffraction patterns (**e**) with simulated diffraction patterns from specific grains of orthorhombic WP. HAADF-STEM image (**f**) and grain orientation map (**g**) of 1D-confined monoclinic WP₂. Each color in a grain orientation map represents a crystal plane corresponding to the color-coded inverse pole figure shown in (**h**). Inverse pole figure for WP₂ (**h**) and electron diffraction patterns (**i**) with simulated diffraction patterns from specific grains of monoclinic WP₂. Scale bar is 1 Å⁻¹. Scale bars: 100 nm for (**b**) and (**f**), 50 nm for (**c**) and (**g**), 1 Å⁻¹ for (**e**) and (**i**). For the overlapping experimental and simulated diffraction patterns, the label provides the nearest zone axis.

Raman spectrum of the WP nanostructures (light green) shows a strong scattering peak at 142.3 cm⁻¹, indicating the $A_g$ vibration mode[15]. At a higher laser intensity (>300 μW), the $A_g$ peak disappeared and the WP nanostructures were oxidized to $WO_3$, as supported by the presence of the Raman peaks (blue) at 130, 264, and 326 cm⁻¹. Thus, we confirm the complete transformation to WP from $WO_2$ at the conversion temperature of 700 °C based on XRD, EDS, and Raman spectroscopy.

**Structure characterization of 1D-WP with varying diameter**

Using transmission electron microscopy (TEM), we characterized the atomic structure of the 1D-confined WP with various diameters. The TEM image presented in Fig. 2a shows the polycrystalline nature of the 1D-confined WP with nanoscale grains. Individual WP grains are merged to form a closed-packed nanowire (width of ≈100 nm) without any pores and noticeable oxide layers. The high-angle annular dark-field scanning transmission electron microscopy (HAADF-STEM) image of one of the grains in the nanowire confirms the atomic structure of WP with bright W atomic columns that arise from the atomic number (Z) difference of $Z_P = 15$ and $Z_W = 74$. The lattice image of the transformed WP agrees with the atomic model viewed along the [001] direction of WP (green: W atoms, grey: P atoms) (Fig. 2b). This is distinguishable from the atomic-resolution HAADF-STEM image of WP₂ and corresponding atomic model (Fig. 2c).

The chemical compositions of the nanowires were analyzed by STEM-EDS (Fig. 2d, e), which showed that a 35 nm-wide nanowire had significantly less P by 63–64% compared to a 250 nm-wide nanowire, which showed roughly 1:1 atomic ratio of W:P. Several nanowires were analyzed using STEM-EDS and showed that below ≈50 nm diameter, the W:P ratio started to deviate from 1:1. This suggests the growth products might contain more than one phase of W-P compounds despite our bulk XRD analysis that only showed the monoclinic WP phase (Fig. 1g). The electron energy loss spectroscopy (EELS) spectra taken from 15-nm-diameter nanowire and 150-nm-diameter nanowire show the P L-edge (Fig. 2f) (for more detailed EELS data, refer to

Supplementary Fig. 6). Micro-structures of two samples with different diameters were examined using HAADF-STEM (≈250 nm diameter in Fig. 2g, h and 15–35 nm diameter in Fig. 2h, j and Supplementary Fig. 7), which showed increasing porosity and more grains with increasing diameter. Surprisingly, the electron diffraction patterns from the 35-nm- and 250-nm-wide nanowires were different, which could not simply be attributed to different tilt angles of the nanowires with respect to the electron beam. STEM-EDS maps and EELS maps for 1D-WP (Fig. 2k, l) and 1D-WP₂ (Fig. 2m, n) clearly show the differences in the size of nanowires.

**Crystallographic orientation mapping of WP and α-WP₂**

To discern additional phases beside WP in the converted phosphides, we carried out 4D-STEM on converted phosphide nanowires to obtain crystal structures of individual grains as illustrated in Fig. 3a (see Methods). With the 4D-STEM method, the electron beam is focused to a nanoscale probe, rastered across the sample surface, and a full diffraction pattern is collected at each spatial coordinate. These diffraction patterns can then be processed to determine the local phase. To do so, we use the ACOM package with py4DSTEM[35,36]. With this analysis method (Supplementary Fig. 8), we compare the experimental diffraction data with simulated diffraction patterns of WP, α-WP₂, and β-WP₂ for all possible orientations, which allows the determination of the crystal phase as well as the crystal orientation. Figure 3b shows a 110-nm-wide tungsten phosphide nanowire; our 4D-STEM analysis indicates that this nanowire is predominantly orthorhombic WP (space group of Pnma). The WP crystal orientation map shows that the nanowire is polycrystalline, with random grain orientations (Fig. 3c, d). Note that for this specimen, the domain size is smaller than the nanowire thickness (in the beam direction), such that all the recorded diffraction patterns contain signals from multiple grains. Accordingly, the orientation maps reflect the orientation of the grain which produces the highest diffraction intensity, and the highest correlation with the simulated patterns (Fig. 3e). In contrast, a 35-nm-wide tungsten phosphide nanowire shown in Fig. 3f was identified to be

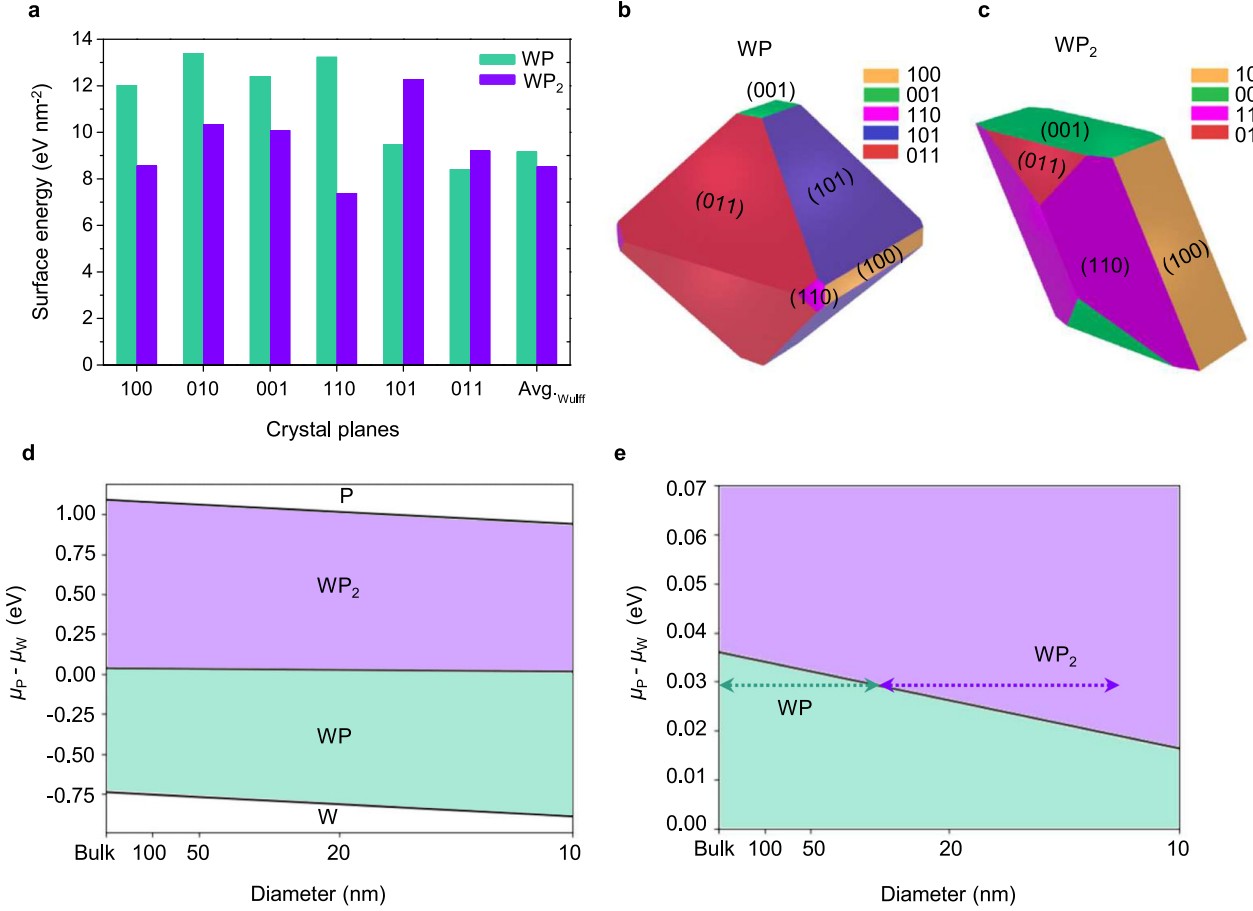

**Fig. 4 | Size-dependent crystallization pathways of WP and α-WP₂. a** Calculated surface energies of various crystal planes for WP (green) and α-WP₂ (purple). Wulff polyhedron constructions for WP (**b**) and WP₂ (**c**) (yellow, 100 plane; green, 001 plane; magenta, 110 plane; blue, 101 plane; red, 011 plane). **d** Phase diagram for tungsten and phosphorus as a function of diameter from bulk to 10 nm. **e** Enlarged phase diagram for tungsten and phosphorus as a function of diameter. Arrows are estimated experimental conditions showing the transition of the phase at ~35 nm.

predominantly monoclinic α-WP₂ (space group of *C2/m*) according to the 4D STEM data and subsequent analysis. This can explain the deviation in the W:P ratio of the nanowires with diameter below ≈50 nm analyzed by TEM-EDS (Fig. 2d), which showed a W:P ratio of ~1:2. The grain orientation map of the α-WP₂ nanowire (Fig. 3g, h) shows random grain orientations (Fig. 3i). Thus, using 4D STEM, we observed diameter-dependent phase bifurcation in the crystallization of 1D-confined tungsten phosphides where WO₂ is converted to WP or WP₂ above and below the diameter range of 35–70 nm, respectively. These two phases for the different diameter nanowires were also confirmed with TEM-EDS (full list of material characterizations and their sample scales for α-WP₂ and WP is provided in Supplementary Fig. 9).

**Size-dependent phosphorization pathways of WP and α-WP₂**

Theoretical calculations were carried out to understand the experimentally observed diameter-dependent phases. We first calculated surface energies of various crystal planes for the two distinct phases of WP and α-WP₂ (Fig. 4a and Supplementary Table 1) and then constructed appropriate Wulff shapes to minimize the total surface energy for each phase (Fig. 4b, c). Since we observed polycrystalline textures for each phase from the grain orientation maps (Fig. 3), the most stable crystallographic planes were averaged. We found that the average surface energy of the α-WP₂ Wulff shape (8.516 eV nm⁻²) is lower than that of the WP Wulff shape (9.162 eV nm⁻²), which suggests that α-WP₂ should be preferred over WP with decreasing diameter that corresponds to increasing surface-to-volume ratio. The surface energy

comparison is consistent with our experimental observation that α-WP₂ is observed in nanowires with diameter below ≈35 nm and WP above ≈35 nm.

To quantify this phase transition, we compute the free energy, $g_{WP}$, of WP and WP₂ phases. In doing so, surface energy becomes a crucial parameter when assuming cylindrical nanowire geometry. We normalize this free energy per atom to allow for a direct comparison between materials, which provides the equation:

$$g_{WP} = E_{Formation,WP} - \frac{1}{2}\mu_{PW} + \frac{4E_{Surface}\upsilon_{WP}}{d} \quad (1)$$

where $E_{Surface}$ is the average surface energy per unit area as calculated from the Wulff model, $\upsilon_{WP}$ is the unit cell volume per atom, $d$ is the nanowire diameter, and $\mu_{PW} = \mu_P - \mu_W$ is the chemical potential difference between P and W. In Eq. (1), the factor of 1/2 corresponds to the mole fraction of P in WP. To obtain the corresponding equation for WP₂, this fraction would simply be modified to 2/3. For both phases, formation energy was taken from the Materials Project database[41]. A phase diagram was calculated as a function of $\mu_{PW}$ and $d$ by equating $g_{WP}$ and $g_{WP2}$ (Fig. 4d, e), which predicts a transition of stable phase from WP to α-WP₂ with decreasing diameter. We note that we had previously studied topological metal MoP nanowires using the same template-assisted growth method[11] and accordingly construct a diameter-dependent phase diagram for MoP (Supplementary Fig. 10), which shows a different behavior from the W-P phase diagram.

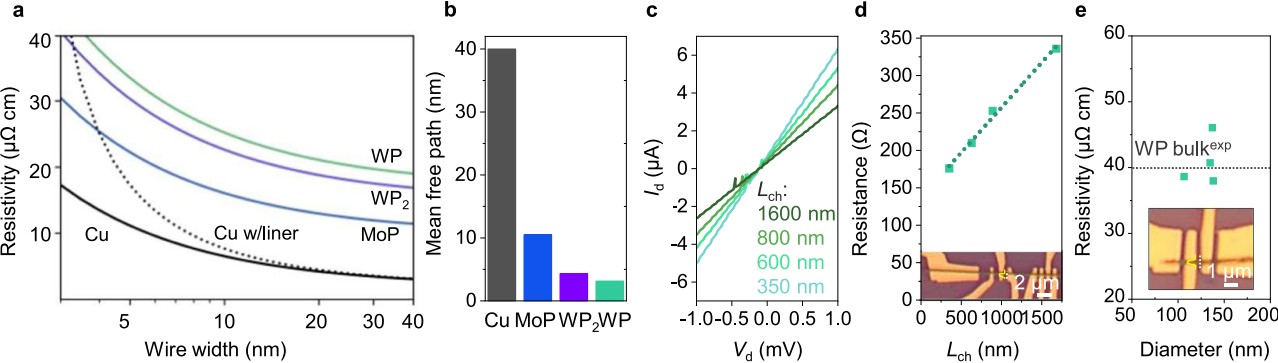

**Fig. 5 | Resistivity scaling of WP. a** Calculated resistivity scaling of topological metal phosphides (WP (green), WP$_2$ (purple), MoP (blue)) as square wires, in comparison with Cu square wires with (dotted black) and without liner (black). The liner was assumed to be 2 nm thick. Single crystal was assumed for the calculations with complete diffuse electron scattering at surfaces ($p = 0$). We note that experimental bulk resistivities ($r_0$) are used when width → ∞. **b** Calculated average mean free paths of Cu (black), MoP (blue), WP$_2$ (purple), and WP (green). **c** Channel-length ($L_{ch}$) dependent current-voltage (I-V) curves of the 1D-confined poly-crystalline WP (turquoise, 350 nm; light green, 600 nm; green, 800 nm; dark green, 1600 nm). **d** $L_{ch}$ dependent resistance variation of the 1D-confined WP. Inset: optical microscopy image and AFM line profile of the measured device. **e** Room-temperature resistivity data of 1D-confined WP with varying cross-sectional area. Dotted line: resistivity value of WP bulk single crystal[16].

## Electron-transport properties of 1D-confined WP

As discussed in the introduction, a subset of TMPs are topological metals that are promising as nanoscaled interconnects. To test the feasibility of these TMPs as interconnects beyond the current 3-nm technology nodes with the metal pitch of ≈24 nm[42], room temperature resistance of these TMPs at the nanoscale must be obtained. Size-dependent room-temperature resistivities of MoP, WP, and WP$_2$ were calculated for two representative geometries of square cross-section wires (Fig. 5a) and thin films (Supplementary Fig. 11) using Fuchs−Sondhemier models to assess electron scattering at surfaces (calculation details in Supplementary Table 2)[43,44]. To minimize the surface scattering, the transport directions were chosen to be along the a- and c-axes for WP and WP$_2$, and the c-axis for MoP given their anisotropic Fermi surfaces (Supplementary Table 3). For comparison, resistivities of Cu wires with and without a liner are also shown in Fig. 5a. For each resistivity curve, complete diffuse surface scattering was assumed with the specularity parameter $p = 0$[45–47]. We did not consider grain-boundary scattering in these calculations because the calculated mean free paths of WP (3.15 nm) and WP$_2$ (4.33 nm) (Fig. 5b) were much smaller than the observed grain sizes (Fig. 3). The dimensional scaling of resistivity for WP and α-WP$_2$ appears similar to that of Cu without the liner and better than Cu with the liner. However, since the bulk resistivity of WP and WP$_2$ is higher than Cu, WP and WP$_2$ do not appear promising as low-resistance interconnects. Further, the resistivity scaling calculations suggest that for WP and WP$_2$, suppressed electron scattering originating from topologically protected surface states is negligible at room temperature, unlike the cases for CoSi and NbAs[23,24].

We carried out room-temperature resistivity measurements on the WP nanowires to experimentally gauge the degree of surface and grain boundary scattering as compared to the calculations. Using the transfer length methods, linear I–V curves were obtained by varying the channel length (Fig. 5c), where the channel resistance increased linearly with the channel length as expected and the contact resistance of 68.9 Ω was extracted (Fig. 5d). Four-probe resistance measurements were carried out on several WP nanowires (Fig. 5e) (two-probe measurement on WP$_2$ nanowire is shown in Supplementary Fig. 12). The resistivity values of our 1D-confined polycrystalline WP, which ranged between 38 and 46 μΩ cm for cross-section areas of 8000–15,000 nm$^2$ (Supplementary Fig. 13), are comparable to that of bulk WP single crystal (40 μΩ cm) grown by chemical vapor transport[16]. Supplementary Fig. 14 shows the correlation between the resistivity and grain structures for WP nanostructures with TEM images. Thus, in agreement with the calculations, the resistivity of 1D-WP implies that electron scattering at grain boundaries is negligible, which can be attributed to the small mean free path (3.15 nm).

## Discussion

We have demonstrated size-dependent phosphorization routes of 1D-confined WP and WP$_2$ from 1D WO$_2$ templates. Crystallographic orientations of grains and grain distributions for each phase were characterized using 4D-STEM, revealing critical transition regime of 35−70 nm diameter below and above which WP$_2$ and WP form. The calculated surface energies from the Wulff constructions of these two competing phases predict α-WP$_2$ to be stable over WP at the nanoscale, which was further shown in the diameter-dependent phase diagrams obtained by DFT calculations. We achieved the lowest resistivity of 38 μΩ cm for WP nanostructures, which is comparable to that of bulk single crystal, indicating electron scattering at grain boundaries and surfaces must be small despite the disordered polycrystalline nature of our samples likely due to their small mean free path. Our findings suggest that the diameter-dependent crystallization route can be exploited to guide synthesis protocols of 1D topological semimetals.

## Methods

### 1D-confined conversion of WP and WP$_2$

1D-WO$_3$ were grown by chemical vapor deposition. WO$_3$ source powder (Sigma-Aldrich, 99.95%) was placed at the center of the hot-walled tube furnace. Chamber pressure was maintained at 3 torr with 20 cm$^3$ STP min$^{-1}$ of H$_2$ gas. C-sapphire substrate with the miscut angle of 1° was placed at upstream of the furnace. The temperature of the upstream was maintained at 600 °C for 10 min. To convert the WO$_2$ templates to WP and WP$_2$, the as-grown 1D WO$_2$ templates were placed in the center of the furnace with 3 g of NaH$_2$PO$_2$·H$_2$O (Sigma-Aldrich, ≥99%) placed upstream. The chamber was pumped down to 100 mtorr, and then 30 cm$^3$ STP min$^{-1}$ of H$_2$ was flowed until the furnace pressure reached atmospheric pressure. The temperature of the furnace was ramped up to 700 °C and held there for 50 min.

### 4D-STEM and EELS measurements

STEM experiments were performed on a $C_s$-probe-corrected Thermo Fisher Scientific Spectra 300 with an extreme-brightness cold field emission gun. The 4D-STEM measurements were collected using an EMPAD at 120 kV and a convergence angle of 0.5 mrad. The 4D-STEM datasets were processed using py4DSTEM and the associated crystal-orientation mapping code[35,36] EELS measurements were conducted using an aberration-corrected TFS Titan Themis 300 X-FEG, equipped

with a Gatan GIF Tridiem energy filter. The microscope was operated at 120 kV, with $a < 100$ pA beam current and a convergence semi-angle of 5 mrad. The stoichiometric ratio of W:P was determined from EDX analysis. From EELS, it is difficult to obtain the composition because of the background subtraction issue. For W, we obtained the W-O edge, which is at 47 eV, since the W-M edge at 1800 eV yields signals that are too weak. The W-O edge is too close to the bulk plasmon, making the power-law-type background subtraction inaccurate. Thus, we cannot obtain the stoichiometric ratio from EELS.

## DFT calculations

All first-principles calculations were performed using the open-source plane-wave software JDFTx[48]. Calculations for all materials were run using the Perdew–Burke–Ernzerhof (PBE) exchange-correlation functional[49] with a plane wave energy cutoff of 680 eV, and ultrasoft pseudopotentials were sourced from the GBRV library[50]. Self-consistent calculations were performed using a Gamma-centered mesh of $12 \times 12 \times 12$ $k$-points to compute the bulk free energies of WP and $WP_2$, whereas surface-oriented slabs utilized a $12 \times 12 \times 1$ $k$-point mesh. Slab geometries were constructed such that the slab thickness was ~30 Å, with a vacuum spacing of 15 Å. A structural relaxation was performed iteratively for bulk and slab structures to optimize the lattice constants and atomic positions. Upon calculating surface energies for WP and $WP_2$, Wulff models were constructed using the WulffPack Python package[51]. For the calculation of bulk resistivities, all electronic structure and phonon properties were transformed into the maximally localized Wannier function basis[52]. In the case of WP, a total of 48 Gaussian Wannier centers were iteratively fitted to the band structure in the energy range $-15.3$ eV to $+4.55$ eV, relative to the VBM, and a phonon $q$-mesh of $4 \times 2 \times 2$ was chosen. For $WP_2$, a total of 48 Gaussian Wannier centers were fitted in the energy range $-16.9$ eV to $+4.38$ eV, and a phonon $q$-mesh of $2 \times 4 \times 2$ was chosen.

## Device fabrication of 1D-confined WP

The converted WP crystals were transferred onto $SiO_2$/Si substrates using a PMMA-assisted wet-transfer method, then coated with e-beam resist layers (200 nm MMA EL 8.5 and 200 nm PMMA A3). Electrode patterns for transfer-length methods and four-probe measurements were written by standard e-beam lithography using a ThermoFisher Helios G4 system. 10/100 nm-thick Cr/Au electrical contacts were deposited by UHV e-beam evaporation followed by in-situ Ar etching (50 W).

## Reporting summary

Further information on research design is available in the Nature Portfolio Reporting Summary linked to this article.

## Data availability

The data that support the findings of this study are available from the corresponding authors upon request.

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

## Acknowledgements

Synthesis and 4D STEM experiments of tungsten phosphides were supported by the Gordon & Betty Moore EPiQS Initiative, grant GBMF9062. Theoretical calculations and transport measurements were supported by the SRC nCORE IMPACT under Task 2966.002 and 2966.005. G.J. acknowledge the additional funding provided by the National R&D Program through the National Research Foundation of Korea (NRF) funded by Ministry of Science and ICT (no. RS-2024-00433166).

## Author contributions

J.J.C. and G.J. supervised the project. G.J. performed synthesis, material characterization, design of growth mechanism, and electrical measurements. C.D.M. and R.S. performed DFT calculations. J.L.H., G.J. N.K.D., H.J.H., and Q.P.S. performed TEM experiments and data analysis. M.T.K., H.W., Y.C., D.J.H. carried out further materials characterization. G.J, C.D.M, J.L.H., H.J.H., R.S. and J.J.C. co-wrote the manuscript. All authors have read the manuscript and commented on it.

## Competing interests

The authors declare no competing interests.
