## [Peer Review File · Nature Communications]

Diameter-dependent phase selectivity in 1D-confined tungsten phosphidesEditorial Note: This manuscript has been previously reviewed at another journal that is not operating a transparent peer review scheme. This document only contains reviewer comments and rebuttal letters for versions considered at *Nature Communications*.

REVIEWER COMMENTS

Reviewer #1 (Remarks to the Author):

Comment: Major Revision

This manuscript mainly focuses on the synthesis of WP and α -WP2, which can be grown via 1D-WO2 templates using the vapor of PH3. The key point is the diameter of the WO2 templates, which may determine the final phase. A phase bifurcation occurs at the cross-over nanowire diameter regime of 35~70 nm. This work provides a novel understanding of 1D-confined topological materials crystallization. However, the performance of the obtained nanowire is not satisfying due to the polycrystallinity. According to the concerns above, I prefer to recommend its publication on Nature Communications after polishing. The suggestions are shown below.

- (1) The atomic-resolution HAADF-STEM images of WP2 should be offered to identify the structure.
- (2) The electron diffraction patterns are the mixture of several series of spots, which are not enough to verify the specific material. The core conclusion needs more supporting information.
- (3) The specific material is identified by comparing the 4D STEM data with simulated diffraction patterns. In Page 9, Line 195, "The grain map of the α -WP2 nanowire (Fig. 3f and 3g) suggests that all the grains are α -WP2." This statement is confusing, which need to modify.
- (4) What is the relationship between grain orientation and surface energy? The Wulff shapes for each phase are different, whether mean different dominant planes?
- (5) This research on WP or WP2 didn't not appear promising low resistance interconnects, which is lack of practical application value. Thus, I think this is where the authors need to solve.

Reviewer #2 (Remarks to the Author):

The manuscript is carefully revised with clearer presentation of data and concept. It is an interesting piece of growth mechanism study of WP system which could provide insights in the general CVT synthesis of phosphide.

It is suitable to publish on Nat. Commun. now.

One further comment is that the author may further consider expanding the introduction with more detailed explanation of related mechanism study, which will make the manuscript's story more straightforward and attractive to a broader audience.

COMMENTS TO AUTHOR:

General remarks and comments of Reviewer #1:

Reviewer #1 (Remarks to the Author):

Comment: Major Revision

*This manuscript mainly focuses on the synthesis of WP and α -WP₂, which can be grown via 1D-WO₂ templates using the vapor of PH₃. The key point is the diameter of the WO₂ templates, which may determine the final phase. A phase bifurcation occurs at the cross-over nanowire diameter regime of 35~70 nm. This work provides a novel understanding of 1D-confined topological materials crystallization. However, the performance of the obtained nanowire is not satisfying due to the poly-crystallinity. According to the concerns above, **I prefer to recommend its publication on Nature Communications after polishing**. The suggestions are shown below.*

We thank the reviewer for the overall positive assessment. Below lists our revisions per the reviewer's suggestions.

(1) *The atomic-resolution HAADF-STEM images of WP₂ should be offered to identify the structure.*

We thank the reviewer for the suggestion. The atomic-resolution HAADF-STEM images of WP₂ were provided in Supplementary Figure 7. In the revision, we have included additional HAADF-STEM images of WP₂ as panel (c) in the revised Figure 2, as shown below. The HAADF-STEM images of WP₂ agree with the expected structure of α WP₂.

Revised Fig. 2 Structural characterization of 1D-confined WP_x ($x = 1$ or 2) with varying diameter.

a, Low-magnification TEM image of 1D-confined WP. **b,c**, Atomic-resolution HAADF-STEM images of WP (**b**) and WP_2 (**c**) obtained from a single grain of the polycrystalline WP and WP_2 nanowires. **d**, STEM-EDS spectra acquired from representative 1D-confined WP_x ($x = 1$ or 2). **e**, Relative atomic content of P with varying diameter, collected from TEM-EDS signals of 7 different samples. **f**, EELS P-edge of 1D-confined WP_x ($x = 1$ or 2). **g,h**, HAADF-STEM images of 1D-confined WP with the diffraction pattern of dominant zone axis of [131] (**g**) and WP_2 with the corresponding diffraction pattern of zone axis of [233] (**h**). **i,j**, False-color STEM-EDS maps (W-L in red, P-K in green) and EELS maps (W-O in red, P-L in green) obtained from two WP nanowires (**i**) and two WP_2 nanowires (**j**). The W:P ratio of the WP nanowire (diameter ~ 796 nm) shown in (**i**) is 0.53:0.47 and the ratio of the WP_2 nanowire (diameter ~ 11 nm) shown in (**j**) is 0.36:0.64.

(2) The electron diffraction patterns are the mixture of several series of spots, which are not enough to verify the specific material. The core conclusion needs more supporting information.

The conclusion of the WP to WP_2 cross-over is not based solely on the electron diffraction patterns. The experimental data that support the WP and WP_2 nanowires and the diameter-dependent crossover are: the STEM-EDS data (Figure 2d,e), atomic-resolution STEM-HAADF images (Figure 2a-c, Supplementary Figure 7), the electron diffraction and 4D STEM data (Figure 2g-h, Figure 3) and Raman data for WP (Supplementary Figure 5). All data consistently support phase pure WP or WP_2 nanowires. Energy calculations (Figure 4) further support the experimental observation. The mixture of several spots in our electron diffraction patterns is due to the polycrystallinity of the nanowires. The 4D STEM analysis (Figure 3) shows clearly that when we get diffraction patterns from individual grains, we can index them to WP or WP_2 .

Here, we provide additional information regarding the 4D-STEM measurements and electron diffraction pattern analysis. In a 4D-STEM experiment, the electron beam is focused to a nanoscale probe, the probe is rastered across the sample, and electron diffraction patterns are collected at every spatial coordinate. These diffraction patterns are then analyzed to determine the local structural phase. To do so, we use the ACOM module within the py4DSTEM package (see refs. 35 and 36 of the main text). With this analysis package, the user inputs candidate crystalline phases to compare against the experimental diffraction data. Based on our STEM-EDS results, we input the crystal structures for WP, α WP_2 , and β WP_2 . The ACOM software then generates diffraction patterns for all possible orientations of the input crystal phases. Each experimental diffraction pattern is then compared with all of the generated patterns (including all orientations of all input crystal files), and the ACOM software calculates the correlation between the experimental dataset and the simulated dataset. The correlation acts as a confidence score that the experimental diffraction pattern actually corresponds to the given crystal structure and orientation.

We add this additional information as new Supplementary Figure 8 in the revision (shown below), using the thin nanowire from Figure 3b as an example. Supplementary Figure 8A shows the virtual STEM-ADF image of the nanowire, and Figure 8B – D shows the ACOM process for α WP₂, β WP₂, and WP, respectively. For each panel, we show the correlation map, which provides the highest correlation for each diffraction pattern, given the input crystal structure. The correlation maps are all shown on the same intensity scale. For the α WP₂ correlation map (Supplementary Fig. 8B), the topmost grain shows a high intensity, indicating a high correlation score, and a high confidence that this grain is α WP₂. We show an example diffraction pattern from this grain, along with the best fit simulated diffraction pattern, which is represented by the overlaid pink circles. The best fit α WP₂ pattern accurately captures the experimental dataset. There are some additional experimental diffraction spots which are not captured by the simulated pattern, but this can be attributed to tails of the STEM probe interacting with neighboring grains. Looking at Supplementary Fig. 8C and D, for the topmost grain, the agreement between experiment and simulation for β WP₂ and WP is poor. This is reflected in the correlation maps: the topmost grain appears dark for both β WP₂ and WP, given the lower correlation. This data indicates that the topmost grain is α WP₂.

Looking at the entire α WP₂ correlation map, the correlation score varies from grain to grain. This can be attributed to a number of factors. First, the simulated diffraction patterns are for defect-free, bulk crystals. Conversely, the studied samples are nanoscale and likely possess some concentration of defects and strain. This will affect the correlation score. Overlapping grains also impact the correlation. Additionally, grains that are close to a zone-axis will have a higher correlation score and be easier to fit relative to grains that are far from a zone axis.

Comparing the correlation maps for the entire nanowire, α WP₂ provides the highest correlation score everywhere except for the bottommost grain. This grain is also marked with an arrow, and the extracted diffraction pattern and ACOM fits are shown. This grain is far from a zone-axis, and the diffraction pattern contains data from an adjacent grain as well, which complicates the analysis. For this grain, the WP provides the highest correlation, followed by α WP₂ and then β WP₂.

In conclusion, the 4D-STEM favors α WP₂ for the vast majority of the nanowire, though there is one grain which is better fit by the WP structure. This is not too surprising, given that the width of this nanowire (~35 nm) is at the transition point from WP to WP₂. Coupling our 4D-STEM data with our atomic-resolution STEM-HAADF and STEM-EDS results, we reaffirm that thin nanowires transition to the α WP₂ phase.

New Supplementary Fig. 8. Virtual 4D-STEM annular dark field (ADF) image of a thin nanowire. **A.** virtual 4D-STEM ADF image of the target nanowire. The blue arrows indicate specific grains which we will use as examples for 4D-STEM ACOM processing. **B.** Fitting to the α WP₂ structure. On the left, we show the correlation map for α WP₂. For each pixel in the map, the diffraction pattern is compared against simulations for α WP₂ at all possible orientations. From all of these comparisons, the best fit (with the highest correlation) is selected. The pixel intensity corresponds to the highest correlation value. To the right, we also repeat the analysis to compare the experimental 4D-STEM data to the simulated patterns of β WP₂ (**C**) and WP (**D**), with the best fit simulations overlap with the pink circles. The intensity scale for the correlation maps shown in **B – D** are the same. The intensities of the correlation maps in C and D are much lower than that shown in B. Combined with our STEM-EDX analysis, we therefore determine that the nanowire is α WP₂.

(3) The specific material is identified by comparing the 4D STEM data with simulated diffraction patterns. In Page 9, Line 195, “The grain map of the α -WP₂ nanowire (Fig. 3f and 3g) suggests that all the grains are α -WP₂.” This statement is confusing, which need to modify.

We thank the reviewer for this comment. To add clarity and better represent our experimental data, we have re-written the entire paragraph including line 195. The updated text is below:

Revisions in the manuscript

(In page 9, line 172)

To discern additional phases beside WP in the converted phosphides, we carried out 4D-STEM on converted phosphide nanowires to obtain crystal structures of individual grains as illustrated in Fig. 3a (details in Methods). With the 4D-STEM method, the electron beam is focused to a nanoscale probe, rastered across the sample surface, and a full diffraction pattern is collected at each spatial coordinate. These diffraction patterns can then be processed to determine the local phase. To do so, we use the ACOM package with py4DSTEM.^{35,36} With this analysis method (Supplementary Fig. 8), we compare the experimental diffraction data with simulated

diffraction patterns of WP, α WP₂, and β WP₂ for all possible orientations, which allows the determination of the crystal phase as well as the crystal orientation. Figure 3b shows a 100 nm-wide tungsten phosphide nanowire; our 4D-STEM analysis indicates that this nanowire is predominantly orthorhombic WP (space group of Pnma). The WP crystal orientation map shows that the nanowire is polycrystalline, with random grain orientations (Fig. 3c and 3d).

(In page 9, line 189)

The grain orientation map of the α -WP₂ nanowire (Fig. 3f and 3g) shows random grain orientations.

(4) What is the relationship between grain orientation and surface energy? The Wulff shapes for each phase are different, whether mean different dominant planes?

The Wulff shapes represent the morphologies that minimize the total surface energy for each phase (WP or α WP₂). Since we experimentally observe that the nanowires are polycrystalline and no preferred grain orientations are detected, we calculate the average surface energy of the Wulff shape for WP and α WP₂, which predicts α WP₂ should be more stable than WP at very small dimensions, in agreement with the experiments.

To quantify this phase transition (*i.e.* At what diameter should the cross over from WP to WP₂ occur?), we compute the free energy, g_{WP} , of WP and WP₂ phases as a function of diameter, assuming cylindrical nanowire geometry, according to equation (1):

$$g_{WP} = E_{Formation_{WP}} - \frac{1}{2}\mu_{PW} + \frac{4E_{Surface}v_{WP}}{d}$$

Here, $E_{surface}$ is the average surface energy per unit area as calculated from the Wulff model, v_{WP} is the unit cell volume per atom, d is the nanowire diameter, and $\mu_{PW} = \mu_P - \mu_W$ is the chemical potential difference between P and W. From this, we can compute the phase diagram, which is shown in Figure 4d and 4e. The trend predicted by the energy calculations agrees with the experimental observation.

In short, the observation that there are no preferred grain orientations allowed us to extract the average surface energies of WP and WP₂ based on the Wulff constructions.

To make this clear, we made the following revision in the main text.

Revisions in the manuscript

(page 11, line 209)

Since we observed polycrystalline textures for each phase from the grain orientation maps (Figure 3), the most stable crystallographic planes were averaged.

(5) This research on WP or WP₂ didn't not appear promising low resistance interconnects, which is lack of practical application value. Thus, I think this is where the authors need to solve.

We acknowledge the reviewer's point regarding the (lack of) potential for tungsten phosphides as on-chip interconnects. We acknowledge these do not appear promising, and our transport data plainly show this.

Our work is the first report of diameter-dependent resistivity scaling on nanoscaled WP. We exploit 1D-confinement as a key factor for the size-dependent selectivity of phase, which has not been reported in previous studies due to the lack of thorough analysis to investigate nanoscaled WP with a few nm resolutions.

Also despite the nanoscale grains, our 1D-confined WP wires exhibit resistivity values comparable to bulk single crystals, indicating minimal grain boundary scattering—a characteristic important for interconnect applications, unlike copper where grain boundary scattering can be responsible for up to 40% of the total increase of the resistivity at the nanoscale (*MRS Bull.* 46, 959-966 (2021)).

The practical impact of our work is that we show WP and WP₂ at the nanoscale are not promising for interconnection applications. This is an important finding for the community so that others do not need to repeat this.

In summary, our work is the first experimental report of the nanoscale resistivity scaling of WP and clear demonstration of phase selectivity due to 1D confinement. And this merits publication irrespective of whether WP or WP₂ fit the interconnection applications.

General remarks and comments of Reviewer #2:

Reviewer #2 (Remarks to the Author):

The manuscript is carefully revised with clearer presentation of data and concept. It is an interesting piece of growth mechanism study of WP system which could provide insights in the general CVT synthesis of phosphide.

It is suitable to publish on Nat. Commun. now.

One further comment is that the author may further consider expanding the introduction with more detailed explanation of related mechanism study, which will make the manuscript's story more straightforward and attractive to a broader audience.

We appreciate the positive feedback from the reviewer on our work. In the revision, we have added extended introduction with a detailed explanation of dimensional conversion mechanisms, particularly emphasizing the influence of nanoscale confinement on crystallization dynamics and phase stability control, as below. For the related mechanism study, we have added two additional references (ref. 13 and 34).

Revisions in the manuscript

(page 3, line 45)

For example, a 2D-confined template is commonly used to achieve transition metal nitrides and transition metal phosphides in recent reports. This method ensures a homogeneous phase across the unconventional 2D structures, regardless of the template's thickness, e.g., MoS₂, TiS₂ or WS₂.^{13, 34} For these crystallization in confined dimensions, atomistic understanding can be further developed to achieve phase selectivity via geometric confinement.

REVIEWERS' COMMENTS

Reviewer #1 (Remarks to the Author):

Comment: Accepted

This work sheds light on the growth mechanism of WP system and provides a new understanding for the crystallization of 1D-confined topological materials. The current manuscript has been carefully revised by the authors and the issues previously raised have been properly addressed, thus I recommend its publication on Nature Communications now.